# Visualization of Runs of Homozygosity and Classification Using Convolutional Neural Networks

**DOI:** 10.3390/biology14040426

**Published:** 2025-04-16

**Authors:** Siroj Bakoev, Maria Kolosova, Timofey Romanets, Faridun Bakoev, Anatoly Kolosov, Elena Romanets, Anna Korobeinikova, Ilona Bakoeva, Vagif Akhmedli, Lyubov Getmantseva

**Affiliations:** 1Faculty of Biotechnology, Don State Agrarian University, Persianovsky 346493, Russia; siroj1@yandex.ru (S.B.); m.leonovaa@mail.ru (M.K.); timofey9258@mail.ru (T.R.); bakoevfaridun@yandex.ru (F.B.); lena9258@mail.ru (E.R.); aaaniich@yandex.ru (A.K.); 2Academy of Biology and Biotechnology Named After D. I. Ivanovsky, Southern Federal University, Rostov-on-Don 344006, Russia; 3All Russian Research Institute of Animal Breeding, Lesnye Polyany 141212, Russia; kolosov777@gmail.com (A.K.); vaqifakhmedli205@mail.ru (V.A.); 4Faculty of Biocybernetics and Systems Biology, Russian State Agrarian University—Moscow Agricultural Academy Named After K. A. Timiryazev, Moscow 127434, Russia; ilonaluba2@mail.ru; 5Faculty of Physics, Mathematics and Natural Sciences, RUDN University: Peoples’ Friendship University of Russia, Moscow 117198, Russia

**Keywords:** runs of homozygosity (ROH), convolutional neural networks (CNN), deep learning, image analysis, genetic architecture, traits, disease, pig, limb defects, breed

## Abstract

In this article, we propose new opportunities for studying ROH and their association with phenotypes. Transforming genetic data into images and analyzing them using CNNs allows us to capture complex homozygosity patterns that are inaccessible through traditional methods. This approach can be applied in animal breeding for accurate breed identification, as well as for studying the relationship between ROH and diseases in both humans and animals (e.g., hereditary defects, resistance to infectious diseases, predisposition to autoimmune disorders, and other complex phenotypes), improving diagnosis and risk prediction by analyzing complex homozygosity patterns in genetic data.

## 1. Introduction

Analysis of Runs of Homozygosity (ROH) is a key tool for understanding the genetic structure and evolution of populations. ROH arise when an individual inherits identical-by-descent (IBD) haplotypes from both parents, reflecting shared ancestry and serving as an indicator of inbreeding levels within a population. The number, length, and genomic distribution of ROH reflect genetic diversity and the demographic history of a population, enabling researchers to study changes in its structure over time [1,2,3,4]. Moreover, ROH are increasingly recognized as a key component of the genetic architecture of complex traits, including productivity and disease resistance in both humans and animals [5,6,7,8,9].

The genomes of livestock species, including pigs, have been shaped by both natural and artificial selection, leading to the emergence of unique genetic patterns, such as ROH [10,11,12,13]. These patterns reflect the history of selection and adaptation processes and may also harbor genes associated with economically important traits [14,15,16]. However, when evaluating the relationship between ROH and complex phenotypes, traditional approaches typically consider parameters such as the total length and count of ROH. While these methods are informative, they do not capture the uniqueness of the ROH landscape, which may be a crucial factor in understanding the genotype–phenotype relationship. Association-based approaches exist; however, selecting the genomic segments to analyze is challenging, and large sample sizes are required, presenting significant obstacles to studying the links between ROH and phenotypes.

Modern high-throughput genomic technologies, in combination with big data analysis methods and deep learning, provide new opportunities for studying ROH. Specifically, transforming genetic data into artificial images enables the visualization of complex patterns such as ROH and the application of computer vision techniques for their analysis. This approach overcomes the limitations of traditional statistical methods, allowing for the consideration of spatial relationships in genetic data that may have previously been overlooked.

Convolutional neural networks (CNNs), which have achieved significant success in computer vision, represent a powerful tool for analyzing genetic data when presented as images. Unlike traditional methods, CNNs can automatically extract complex patterns and perform dimensionality reduction, making them particularly effective for classification and recognition tasks [17,18,19,20,21,22]. The success of CNN-based computer vision systems has driven major technology companies, as well as a rapidly growing number of startups, to invest in research and development initiatives focused on image recognition, including applications in healthcare and agriculture. However, the direct use of genomic representations as images and their subsequent analysis using computer vision remain relatively uncommon in genotype–phenotype and disease-related studies. Instead, such studies typically emphasize the transformation of genomic sequence data, transcriptome analysis, and other “omics” data.

The potential of deep learning for genome sequence classification, particularly in exon–intron analysis and related fields, has been demonstrated in the work of Ben Nasr Barber et al. [23]. An intriguing image-based approach to genetic data analysis was introduced by Chen et al. [24], who aimed to classify schizophrenia using genomic data, specifically single nucleotide variants (SNVs) identified through genome-wide association studies (GWAS) and deep learning methods. To generate images, SNVs were transformed into heat maps (indicating presence or absence), fixed-size matrices, and graph-based representations (visualizing genetic networks or linkage disequilibrium).

In another study, Kupperman et al. [25] proposed a real-time HIV outbreak detection system based on genetic data. Visual representations, such as phylogenetic trees and sequence similarity matrices, were utilized as input data to enhance model interpretability. Similarly, Mahin et al. [26] investigated a novel approach for classifying single-cell RNA sequencing data, using gene expression profiles as image inputs.

Overall, converting genetic data into images for subsequent analysis using CNNs holds significant promise for research involving multidimensional biological data. However, the application of this approach to analyzing runs of homozygosity (ROH) and their relationship with phenotypic traits has not yet been explored.

Inspired by deep learning advancements in image analysis, we propose a novel strategy for studying ROH through the creation of ROH maps and their analysis using CNNs. This strategy consists of three main stages: (1) ROH detection, (2) visualization and mapping of ROH for each individual, and (3) classification and differentiation of samples based on their unique ROH architecture. This approach not only allows for the assessment of inbreeding and genetic diversity but also enables the determination of breed affiliation and the relationship between ROH and complex traits. The combination of modern genomic technologies, machine learning methods, and computer vision opens up new possibilities for studying ROH and their role in the genetic architecture of complex traits.

## 2. Materials and Methods

### 2.1. Animals and Traits

The research design was structured as follows. In the first stage, the goal was to determine the breed of pigs by analyzing ROH maps. Genotyping data from Large White (LW, n = 568) and Duroc (DR, n = 600) pigs were used. ROH regions were identified within each breed, and ROH maps were constructed for each individual. These maps were then used to train a convolutional neural network (CNN) to classify pigs by breed.

In the second stage, the goal was to identify binary traits in pigs of the same breed based on ROH maps. The selected trait was the presence or absence of defects (bumps/growths) on the hind limbs. Two groups were formed: Large White pigs without defects (LW1, n = 364) and those with defects (LW2, n = 204). ROH maps were generated for each individual, and a CNN was trained to predict the presence or absence of defects. The phenotype “bumps/growths” refers to small bumps or growths in the hock joint area of the hind limbs [27,28]. These were recorded by farm specialists during routine pig inspections.

### 2.2. Genotyping and Quality Control

DNA extraction was performed by farm specialists, who collected tissue samples from Large White pigs in accordance with standard monitoring procedures and farm guidelines. No procedures requiring ethical approval were conducted, as tissue samples (plucks) were obtained as part of routine breeding practices.

Genotyping was carried out using the GeneSeek^®^ GGP SNP80x1_XT (Illumina Inc., San Diego, CA, USA). Quality control (QC) was performed using PLINK v1.9 [29], excluding individuals with more than 10% missing genotypes (—mind 0.1) and SNPs with more than 10% missing values (—geno 0.1). Genotypic data for Duroc pigs were obtained from the open scientific repository Figshare [30].

### 2.3. Analysis of Homozygosity (ROH)

ROH detection was performed using PLINK v1.9 with the following parameters: minimum number of SNPs in a homozygous segment (—homozyg-snp 50), minimum length of a homozygous segment (—homozyg-kb 1000), minimum density (—homozyg-density 100), maximum distance between two SNPs from different segments (—homozyg-gap 500), number of SNPs in a sliding window (—homozyg-window-snp 50), maximum number of heterozygous SNPs in a sliding window (—homozyg-window-het 1), and maximum number of heterozygous SNPs in an entire ROH segment (—homozyg-het 1).

ROH segments were classified based on their length (Mb) into the following categories: less than 2 Mb, 2–4 Mb, 4–6 Mb, 6–8 Mb, 8–16 Mb, and more than 16 Mb.

### 2.4. Drawing Maps

ROH maps were generated using a slightly modified version of the roh_visual function from the HandyCNV [31] package in the R environment (The script used for image generation is included in Appendix A). Each segment was represented by a different color: green (less than 2 Mb), red (2–4 Mb), blue (4–6 Mb), turquoise (6–8 Mb), orange-yellow (8–16 Mb), and gray (more than 16 Mb). Maps were created for each animal and saved in JPEG format with a resolution of 512 × 512 pixels.

### 2.5. Convolutional Neural Network (CNN) Model

For model training, the data were divided into three subsets: training, validation, and test sets, with a 70:20:10 ratio, respectively. The split was performed using a stratified approach to maintain class proportions in each subset. Additionally, image label data were converted into one-hot encoding format, allowing them to be used as target variables in the neural network. For image processing, a classical convolutional neural network (CNN) architecture was employed, which included several convolutional, normalization, and pooling layers. Dropout regularization was applied to prevent overfitting. Model evaluation was performed using metrics such as accuracy, sensitivity, specificity, positive predictive value, and negative predictive value.

### 2.6. Cross-Validation and Comparison with Classical Methods

To assess the robustness of the CNN model, we employed 10-fold cross-validation. The dataset was partitioned into 10 subsets, with each subset serving as the validation set in turn, while the remaining subsets were used for training. The model was trained for 100 epochs per fold, and performance metrics were recorded accordingly. For each fold, a model was instantiated using create_model (), trained for 100 epochs (epochs = 100), and the training history along with the evaluation metrics was stored in the cv_results list.

For comparison with classical machine learning approaches, logistic regression was applied to features derived via dimensionality reduction using Principal Component Analysis (PCA). The image data were flattened into one-dimensional vectors, and PCA was implemented using the robust SVD algorithm (RSpectra::svds). The logistic regression model was likewise evaluated using 10-fold cross-validation, with class balancing achieved through an upsampling strategy. Model performance was assessed using the ROC-AUC metric (twoClassSummary).

### 2.7. Identifying Informative Regions

To interpret the CNN results and identify informative regions within the ROH maps, we employed Saliency Maps [32,33,34]. This method provides fine-grained insights by quantifying the influence of individual input pixels on the model’s predictions. Unlike techniques that broadly highlight regions of importance, Saliency Maps enable the precise localization of pixel-level variations that have the greatest impact on classification outcomes.

## 3. Results

### 3.1. Classification by Breed in Pigs

After combining the genotyping data of Large White and Duroc pigs and applying filters, a total of 25,105 variants were retained. Scanning for runs of homozygosity (ROH) revealed 14,862 segments in Large White (LW) pigs, averaging 26 segments per animal, and 25,660 segments in Duroc pigs, averaging 42 segments per animal. Summary statistics for the segments are provided in the Appendix A. The distribution of segments by length (<2 Mb, 2–4 Mb, 4–6 Mb, 6–8 Mb, 8–16 Mb, and >16 Mb) is presented in Figure 1.

Following the identification and length-based classification of the ROH segments, ROH maps were created for each animal. Figure 2 shows the ROH map for a representative animal. Each row in the figure corresponds to a chromosome, arranged from top to bottom, from chromosome 1 to 18. Chromosome numbers are not displayed in the figure to avoid noise during image recognition. The ROH segments are classified by length in Mb and are represented by different colors: segments less than 2 Mb are green, 2–4 Mb are red, 4–6 Mb are blue, 6–8 Mb are turquoise, 8–16 Mb are orange-yellow, and segments longer than 16 Mb are gray.

Based on the visualized ROH maps, we developed a model to determine the inter-breed differentiation of Large White and Duroc pigs. The model achieved perfect scores across all metrics, including accuracy, sensitivity, specificity, positive predictive value, and negative predictive value, all equaling 1.0. This indicates flawless model performance and 100% accuracy in determining the breed of pigs based on the genomic landscape of ROH segments (Table 1, Appendix A).

To evaluate the model’s robustness, we performed 10-fold cross-validation, which confirmed the high reproducibility of the results. A comparison between the CNN and logistic regression models is provided in Appendix A. Both individual and averaged saliency maps, generated using the Saliency Map method, are also presented in Appendix A.

### 3.2. Binary Trait Classification in Large White Pigs

For the classification of ROH maps based on a binary trait within the Large White breed, two groups were created: LW1 (absence of limb defects) and LW2 (presence of limb defects). After quality control of the genotyping data, 54,691 variants were retained. Scanning for regions of homozygosity (ROH) revealed 31,141 segments in LW1 (averaging 85 segments per animal) and 17,913 segments in LW2 (averaging 87 segments per animal). Summary statistics for the segments are provided in the Appendix A. The distribution of segments by length (<2 Mb, 2–4 Mb, 4–6 Mb, 6–8 Mb, 8–16 Mb, and >16 Mb) is presented in Figure 3.

Based on the visualized ROH maps, we developed a convolutional neural network (CNN) model to determine the presence or absence of defects in pigs. The model demonstrated moderate accuracy (78.57%, 95% CI: 0.6319–0.897), indicating its ability to distinguish between animals with and without defects based on the ROH maps (Table 2, Appendix A).

The sensitivity (73.33%) and specificity (81.48%) of the model are at an acceptable level, suggesting that it can effectively identify both positive and negative classes. It is important to note that the defects in question—manifesting as small bumps/growths around the hock joint in the hind limbs of pigs—are a complex multifactorial process influenced by both genetics and external conditions. Consequently, the relatively low positive predictive value (68.75%) suggests the presence of additional factors affecting the phenotype. However, the high negative predictive value (84.62%) indicates that the model confidently identifies animals without defects, making it useful for excluding individuals that do not have a predisposition to the condition. Overall, the results demonstrate that the analysis of ROH maps using CNN allows for the determination of genetic predisposition to the development of bumps/growths in pigs.

### 3.3. Ensuring Effective Model Operation

*Data Preparation*. Images are preprocessed and loaded into memory for random sample generation. The model accepts input images of size 512 × 512 pixels with three channels (RGB), ensuring standard color image representation. Each input image is passed through the layer input layer, which defines the input shape as shape = c(512, 512, 3), specifying height, width, and color channels. To ensure consistency in data representation, pixel values are normalized by scaling them to the range [0, 1] (using 1/255). The dataset is then divided into two subsets: the training set, used for model learning, and the validation set, used to assess model accuracy.

*Deep Learning Model Development*. The model architecture consists of convolutional layers, max-pooling layers, a global average pooling layer, and dense layers. Feature extraction is performed through convolutional layers: the first layer applies 8 filters (3 × 3) to detect local features, the second layer increases to 16 filters, allowing the model to learn more complex patterns, and the third expands to 32 filters, enabling the identification of higher-level abstractions. 

Activation and Regularization. To introduce non-linearity, the ReLU activation function is applied, while L2 regularization is used to prevent overfitting by controlling weight magnitudes. 

*Normalization and Pooling.* Batch normalization is incorporated to accelerate training and stabilize outputs, mitigating issues like vanishing gradients. Max pooling is applied to reduce the dimensionality of extracted features, lowering computational costs while preserving essential information. 

Final Classification Layer. The last layer consists of a fully connected (dense) layer with two neurons, one for each class. A Sigmoid activation function is used to convert the output into class probabilities.

*Model Training*. The model is trained for 100 epochs, balancing the risk of underfitting and overfitting. A batch size is defined, and class weights are applied to handle class imbalance. Binary cross-entropy loss with sigmoid activation is used for classification. Additionally, focal loss can be applied to improve performance on imbalanced datasets by emphasizing hard-to-classify samples. Early stopping is implemented to prevent overfitting by automatically halting training when no further improvement is observed on the validation set.

*Model Evaluation.* Model evaluation involves generating labels for the analyzed data and assessing the model’s performance based on metrics such as accuracy, sensitivity (recall), specificity, positive predictive value (PPV), and negative predictive value (NPV), using test data that were not involved in the training or validation processes.

## 4. Discussion

In this study, we proposed a novel approach to analyzing runs of homozygosity (ROH) using convolutional neural networks (CNNs), which enables the visualization and classification of genetic data as images. This method opens new perspectives for studying the genetic structure of populations and applying these data in agriculture and medicine. ROH maps are created by transforming genetic data into visual images, where each ROH segment is represented as a colored area on a chromosome. This allows us to observe complex patterns of homozygosity that are difficult to interpret from numerical data. CNNs are capable of automatically detecting complex patterns in images, making them an ideal tool for analyzing ROH maps. Unlike traditional statistical methods, CNNs can consider not only individual segments but also their spatial relationships, revealing hidden patterns.

In this work, we demonstrated that the visualization of ROH and analysis using CNNs successfully classify pigs of different breeds. This indicates that the ROH patterns, visualized as images, contain sufficient information for accurate breed determination. The visualization and classification of ROH architecture opens new opportunities for studying demographic history and breed-specific genetic structures. This can be used to identify genetic differences between breeds and offers new perspectives in animal selection, the creation of new breeds, and the optimization of hybridization systems in pig farming.

Our results demonstrate that converting genetic data into images and analyzing them with CNNs can be an effective tool for studying the relationships between genetic ROH patterns and complex phenotypic traits. The binary trait classification model showed the ability to determine the presence or absence of defects (bumps/growths) in pigs with 78.57% accuracy. Although this result is lower than in the case of breed classification, it still demonstrates the potential of using ROH for analyzing complex phenotypic traits. Given that such defects may be multifactorial and depend on both genetics and external conditions, the achieved accuracy is encouraging.

### Limitations and Prospects

Various programs, including RZooROH [35] and PLINK [36], exist for detecting ROH, each with its own methods for identifying these regions. RZooROH uses a Bayesian approach to full probabilistic modeling to determine regions of homozygosity. PLINK uses a genome scanning method with a sliding window. If a certain proportion of consecutive SNPs are homozygous (above a predetermined threshold, usually 90–100%), that region is considered ROH.

Identifying ROH using PLINK is one of the most widely used tools for genomic analysis because it is fast and efficiently detects ROH while handling various types of genomic data [37]. For detecting ROH to build a homozygosity map, we preferred PLINK due to its simplicity and speed. However, a limitation of this program is that it does not distinguish homozygosity by descent from homozygosity by state—it identifies any long homozygous segments regardless of whether they originated from a common ancestor. Therefore, how the model’s accuracy, especially for complex traits, changes with different ROH identification methods requires further study.

Additionally, attention should be paid to the original genomic data. We used genotyping data from chips. In the first case, combining two datasets left us with 25,105 variants, based on which the model performed excellently in classifying breeds. However, the number of variants significantly influences the number of identified ROH segments. For Large White pigs, 14,862 segments were identified based on 25,105 variants (an average of 26 per animal), while 48,848 segments were identified using 54,691 variants (86 per animal). The main difference in the number of identified ROH segments is due to short segments (1 to 2 Mb). Short ROH segments may not significantly affect the overall genetic structure of modern pig breeds, especially if they do not contain functionally significant genes. However, for complex traits such as disease resistance or productivity, accurately identifying all ROH segments can be critical, as even short segments may contain key genes or regulatory elements. In this case, sequencing data (whole-genome sequencing, WGS) allows for more accurate identification of ROH, including short segments that may be missed when using chips. Using sequencing data can improve the accuracy of CNN models, as they will work with more detailed ROH maps.

Furthermore, the models were not evaluated using an independent external dataset, which limits the ability to draw conclusions about their generalizability to different population groups. In the future, testing on broader and more diverse datasets will help optimize the models and ensure the robustness and applicability of the proposed approach.

## 5. Conclusions

The proposed approach of visualizing homozygosity segments and classifying them using convolutional neural networks demonstrates significant potential for analyzing ROH and their associations with phenotypes. Converting genetic data into images and analyzing them using CNNs allows for the consideration of complex homozygosity patterns that are inaccessible to traditional methods. Saliency maps represent a promising tool for identifying informative regions within ROH maps by quantifying the pixel-level influence on model predictions. This facilitates improved interpretability through the precise localization of homozygosity patterns relevant to the target phenotypes. This method can be applied not only in agriculture but also in medicine to study the relationship between ROH and diseases. In the future, we plan to expand the application of this approach to analyze other types of genetic data and their associations with phenotypes, opening new perspectives for research in genetics and selection.

## Figures and Tables

**Figure 1 biology-14-00426-f001:**
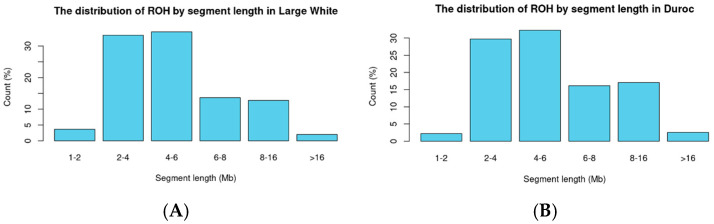
The distribution of ROH by segment length ((**A**) The distribution of ROH by segment length in Large White; (**B**) The distribution of ROH by segment length in Duroc).

**Figure 2 biology-14-00426-f002:**
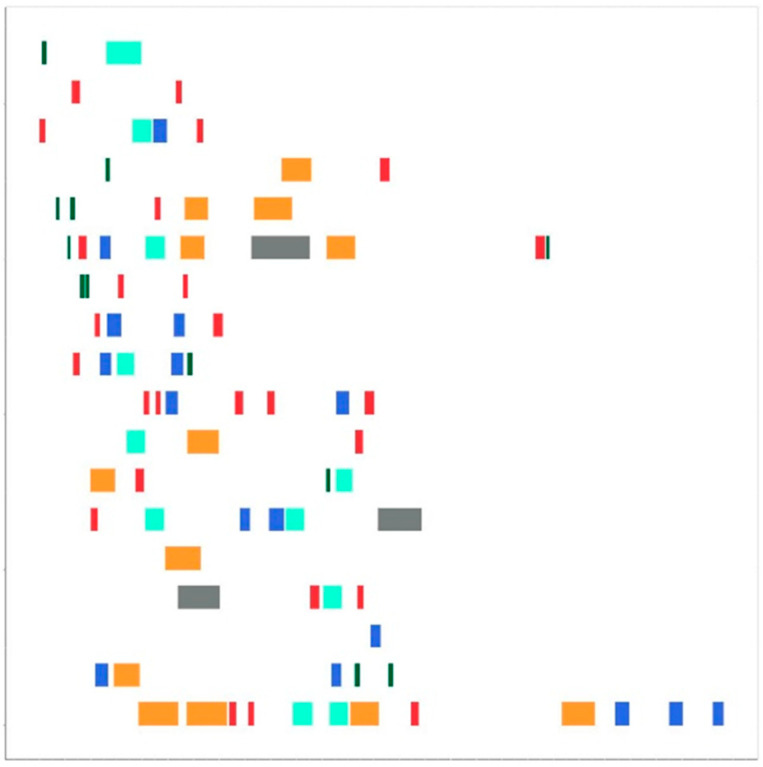
ROH map for one animal.

**Figure 3 biology-14-00426-f003:**
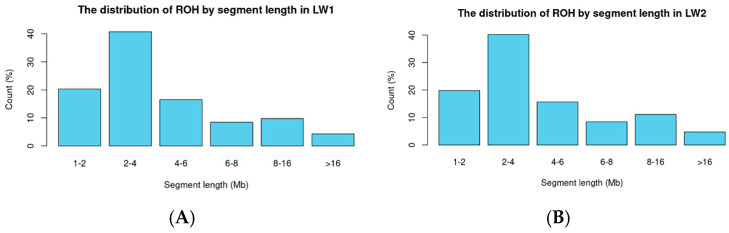
The distribution of ROH by segment length ((**A**) The distribution of ROH by segment length in LW1; (**B**) The distribution of ROH by segment length in LW2).

**Table 1 biology-14-00426-t001:** Confusion matrix for the inter-breed differentiation model based on ROH maps.

	Observed Values			
Predicted Values	LW	D		
LW	30	0		
D	0	37		
Accuracy	Sensitivity	Specificity	Positive Predictive Value	Negative Predictive Value
1.0 (95% CI (0.95))	1.0	1.0	1.0	1.0

**Table 2 biology-14-00426-t002:** Confusion matrix for the binary trait model in Large White pigs based on ROH maps.

	Observed Values			
Predicted Values	LW1	LW2		
LW1	22	4		
LW2	5	11		
Accuracy	Sensitivity	Specificity	Positive Predictive Value	Negative Predictive Value
0.7857 95% CI: (0.6319, 0.897)	0.7333	0.8148	0.6875	0.8462

## Data Availability

Data are available upon reasonable request. All scripts used in this study are available upon request from the authors.

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
