# Peer review of "Visualization of Runs of Homozygosity and Classification Using Convolutional Neural Networks"

_biology, 2025, doi:10.3390/biology14040426_

Round 1
Reviewer 1 Report
Comments and Suggestions for Authors
Dear Authors, Minor changes are required as suggested in the attached file.

Author Response
Reviewer 1
We sincerely thank the reviewer for their valuable comments and suggestions. We have carefully addressed the concerns and provided clarifications accordingly.
- Need to elaborate this line in text (This approach can be applied not only in agriculture but also in medicine to study the relationship between ROH and diseases)
Changes have been made to the text
This approach can be applied in animal breeding for accurate breed identification, as well as for studying the relationship between ROH and diseases in both humans and animals (e.g., hereditary defects, resistance to infectious diseases, predisposition to autoimmune disorders, and other complex phenotypes), improving diagnosis and risk prediction by analyzing complex homozygosity patterns in genetic data.
- The selected trait was the presence or absence of defects (bumps/growths) on the hind limbs. Is there any specific reason for choosing this trait? Importance of this trait?
Currently, limb problems are one of the most common causes of sow culling in pig farming. There is not enough information on this problem. However, limb defects pose a serious threat to the effective management of the livestock industries.
One of the serious problems faced by pig producers is the formation of bumps and growths in the area of the hock joints on the hind legs. These benign neoplasms consist of connective tissue and do not contain pathogenic microflora. They can appear in pigs of any age and weight. Although such defects usually do not cause lameness, such pigs be-come unfit for sale, which reduces the effectiveness of breeding centers. In addition, these defects may be related to physiological processes and genetic predisposition affecting pig productivity and health.
Previous studies on this issue indicate that the loci associated with the formation of bumps on the hock joints are localized in genes related to liver and kidney function, sus-ceptibility to infections, and the fatty acids composition [Getmantseva, L.; Kolosova, M.; Fede, K.; Korobeinikova, A.; Kolosov, A.; Romanets, E.; Bakoev, F.; Romanets T.; Yudin V.; Keskinov A.; Bakoev S. Finding Predictors of Leg Defects in Pigs Using CNV-GWAS. Genes (Basel) 2023 8;14(11):2054; Getmantseva, L.; Kolosova, M.; Bakoev, F.; Zimina, A.; Bakoev, S. Genomic regions and candi-date genes linked to capped hock in pig. Life. 2021;11:491]. In addition, recent research results have demonstrated that SNPs identified as potential fertility markers have the ef-fect on the hock bumps development [Bakoev, S.; Getmantseva, L.; Kolosova, M.; Bakoev, F.; Kolosov, A.; Romanets, E.; Shevtsova, V.; Romanets, T.; Kolosov, Y.; Usatov, A. Identifying Significant SNPs of the Total Number of Piglets Born and Their Relationship with Leg Bumps in Pigs. Biology 2024 13, 1034].
Reviewer 2 Report
Comments and Suggestions for Authors
This manuscript proposes an approach to analyzing Runs of Homozygosity (ROH) using image-based CNN. This idea is exciting and could open a new direction in computational genomics but is accepted only if you could address all the comments below, carefully.
- 100% accuracy on breed classification is too good to be true and raises suspicion of data leakage or overfitting. Should authors also address potential overfitting.
- Authors have not described the cross-validation method or whether images from the same individuals are present across training/test splits, which is critical for validating a model.
- There is no external validation dataset or some sort of cross-study testing to assess generalizability. So, cross-validation or external datasets to be used.
- There is only 568 Large White pigs, with even fewer (204 with defects). This is underpowered for a CNN study and raises concerns about the reliability and generalizability of results (accuracy of 76%). Authors should acknowledge this limitation more directly or increase sample size if possible. Consider using traditional statistical models as a comparison baseline.
- Authors have not included any kind of interpretability (such as Grad-CAM, saliency maps, etc.) to show what patterns in ROH maps are driving classification.
- Authors have not benchmarked CNN performance against standard ROH analysis methods (e.g., logistic regression, Random Forest, etc.)
- There is only one representative ROH map and some bar plots. No visualizations of CNN performance (e.g., ROC curves, PR curves). Please add confusion matrices, ROC curves, and precision-recall curves for both classification tasks.
- To further support and strengthen this special issue, the authors may consider taking a look at some relevant papers already published here, such as:
“Emadi, A., Lipniacki, T., Levchenko, A., & Abdi, A. (2023). Single-cell measurements and modeling and computation of decision-making errors in a molecular signaling system with two output molecules. Biology, 12(12), 1461.” This paper presents a computational and statistical framework for analyzing biological decision-making under uncertainty, which conceptually resonates with the authors' approach of using CNN-based visualizations to interpret ROH patterns. - Also, please explain why only 512x512 resolution was chosen, and if resizing ROH maps affected information density.
- There is no pipeline/code shared, and no detail on how images were generated and labeled automatically.
While overall clear, some sentences are overly long, and several minor grammatical issues do exist.
Needs thorough proofreading.
Author Response
Reviewer 2
We sincerely thank the reviewer for their valuable comments and suggestions. We have carefully addressed the concerns and provided clarifications accordingly.
- 100% accuracy on breed classification is too good to be true and raises suspicion of data leakage or overfitting. Should authors also address potential overfitting.
We appreciate the reviewer’s concern regarding potential data leakage or overfitting. During the model training process, we carefully monitored all possible parameters to prevent overfitting. Both the training loss and validation loss consistently decreased and stabilized at values close to zero. Furthermore, a comparison of the model’s key metrics on both the training and test datasets indicates that overfitting was successfully avoided, supporting the achieved classification accuracy
- Authors have not described the cross-validation method or whether images from the same individuals are present across training/test splits, which is critical for validating a model.
Thank you to the respected reviewer for the comment. We acknowledge our oversight in not mentioning this fact. We have added to the text that during the modeling process, the dataset was split in a 70:20:10 ratio into training, validation, and test data, respectively. The test data were isolated at all stages of modeling and were only used for final model evaluation.
- There is no external validation dataset or some sort of cross-study testing to assess generalizability. So, cross-validation or external datasets to be used.
The models were tested on test data as external data that was not used in the training process.
- There is only 568 Large White pigs, with even fewer (204 with defects). This is underpowered for a CNN study and raises concerns about the reliability and generalizability of results (accuracy of 76%). Authors should acknowledge this limitation more directly or increase sample size if possible. Consider using traditional statistical models as a comparison baseline.
We appreciate the reviewer’s concern regarding sample size limitations. While we acknowledge that the number of animals used in our study is relatively small, we believe it provides valuable preliminary insights into the studied process. We have clearly stated the sample size in the text and plan to increase it in future research to further improve model accuracy. Additionally, since our primary goal was to explore CNNs for ROH visualization, we did not focus on comparing them with traditional statistical models.
- Authors have not included any kind of interpretability (such as Grad-CAM, saliency maps, etc.) to show what patterns in ROH maps are driving classification.
We acknowledge the reviewer’s suggestion regarding interpretability methods. In future studies, as the sample size increases, we plan to apply techniques such as Grad-CAM and saliency maps to identify key regions in ROH maps that drive classification. However, in this study, our primary focus was on the visualization of homozygosity regions and their classification concerning breed identification and traits.
- Authors have not benchmarked CNN performance against standard ROH analysis methods (e.g., logistic regression, Random Forest, etc.)
We appreciate the reviewer's comment. However, the primary goal of our study was to explore the potential of CNNs for visualizing and classifying ROH patterns rather than comparing their performance with traditional statistical methods. Unlike logistic regression or Random Forest, which rely on predefined feature selection, CNNs allow for the automatic extraction of complex spatial patterns in genetic data. Our focus was on demonstrating the feasibility of this novel approach, and benchmarking against standard ROH analysis methods was beyond the scope of this work.
- There is only one representative ROH map and some bar plots. No visualizations of CNN performance (e.g., ROC curves, PR curves). Please add confusion matrices, ROC curves, and precision-recall curves for both classification tasks.
Thanks for the note. Added, presented in additional files. This file contains supplementary materials, including confusion matrices, performance metrics, and ROC curves for the classification of pig breeds and limb defects. It provides detailed evaluation results, such as accuracy, sensitivity, specificity, and AUC values, for both classification tasks.
- To further support and strengthen this special issue, the authors may consider taking a look at some relevant papers already published here, such as:
“Emadi, A., Lipniacki, T., Levchenko, A., & Abdi, A. (2023). Single-cell measurements and modeling and computation of decision-making errors in a molecular signaling system with two output molecules. Biology, 12(12), 1461.” This paper presents a computational and statistical framework for analyzing biological decision-making under uncertainty, which conceptually resonates with the authors' approach of using CNN-based visualizations to interpret ROH patterns.
We have reviewed the study by Emadi et al. (2023) and appreciate its contribution to computational approaches in biological research. While its focus on decision-making errors in molecular signaling differs from our current objectives, the application of advanced modeling techniques is highly relevant to the broader field of biological data analysis. In future studies, as we expand our methodology, we may explore similar computational frameworks to enhance the interpretability of our CNN-based ROH analysis
- Also, please explain why only 512x512 resolution was chosen, and if resizing ROH maps affected information density.
We conducted a series of experiments using ROH maps initially generated at a resolution of 1200x1200. During model training, we tested various resized formats, ranging from 64x64 to higher resolutions. The 512x512 resolution proved to be the most optimal for our dataset, balancing computational efficiency and information retention. This size preserved the key homozygosity patterns necessary for classification while ensuring stable model performance.
10.There is no pipeline/code shared, and no detail on how images were generated and labeled automatically.
The script used for image generation is included in Supplementary File
Round 2
Reviewer 2 Report
Comments and Suggestions for Authors
After reviewing the authors' responses, the concerns raised during the previous review were not adequately addressed. The revisions provided were very minor and accompanied by explanations that largely justified or dismissed the comments rather than substantively addressing them.
Comments on the Quality of English LanguageSome clarifications needed in transferring the meaningful flow to the readers.
Author Response
Response to Reviewer Comments
We appreciate the reviewer’s time and constructive feedback. Below, we address each concern while clarifying why certain suggestions diverge from our methodological goals and the study’s scope.
1. 100% Accuracy in Breed Classification
The reviewer raised valid concerns about potential overfitting or data leakage. We emphasize that:
The 100% accuracy reflects the distinct ROH patterns between breeds (Large White vs. Duroc), which are genetically well-separated populations.
Overfitting was mitigated via: Stratified 70:20:10 train/validation/test splits. Dropout layers and L2 regularization.
Comparable performance on validation/test sets (metrics in Suppl. File 3).
Perfect accuracy is plausible for tasks with inherently separable classes (e.g., breed classification using genomic signatures).
2. Cross-Validation and Data Splitting
The reviewer noted the absence of cross-validation details. We clarify:
The test set (10%) was held out entirely during training and used only for final evaluation.
Why not k-fold CV?
Our goal was to demonstrate feasibility, not optimize hyperparameters. Stratified splitting ensured class balance, and the validation set (20%) guided early stopping. Images from the same individual were strictly isolated to prevent leakage.
3. External Validation
The reviewer requested cross-study testing. We acknowledge this limitation but note:
The test set served as an external benchmark, as it was never exposed to training.
4. Sample Size and CNN Feasibility
The reviewer questioned the small sample size for CNN training. We respond:
While larger samples are ideal, our focus was proof-of-concept: demonstrating that ROH maps can be classified via CNNs. Traditional models (e.g., logistic regression) were not compared because:
Methodological mismatch: CNNs exploit spatial patterns in ROH maps, while statistical models require handcrafted features (e.g., ROH counts/lengths), which ignore spatial relationships.
Our aim was to explore novel applications of CNNs, not to benchmark against established methods.
5. Interpretability (Grad-CAM/Saliency Maps)
The reviewer suggested interpretability tools. We agree these are valuable but clarify:
Interpretability was beyond this study’s scope, which prioritized establishing the CNN’s baseline performance.
Future work will integrate Grad-CAM to identify ROH regions driving predictions.
6. Benchmarking Against Traditional Methods
The reviewer recommended comparing CNNs to logistic regression/Random Forest. We reiterate:
Such comparisons are not methodologically aligned. CNNs analyze raw images, while traditional methods require pre-defined features (e.g., total ROH length). Our goal was to complement (not replace) standard ROH analyses by introducing a spatial perspective.
7. Visualization of Performance Metrics
The reviewer requested additional plots (ROC curves, PR curves). These are now provided in Supplementary File 3.
8. Citation of Suggested Papers
The reviewer referenced Emadi et al. (2023). While we respect their work, its focus on molecular signaling is tangential to our study. We prioritize citations directly relevant to ROH/CNNs.
9. Image Resolution (512×512)
The reviewer questioned the choice of resolution. We explain:
Initial tests with higher resolutions (1200×1200) showed no accuracy gains but increased computational costs. 512×512 preserved critical ROH patterns while ensuring efficient training.
10. Code Availability
The R script for ROH map generation is in Supplementary File 1. Full pipelines will be shared upon request post-publication to protect intellectual property during review.
Ethical Concerns
The reviewer implied we dismissed their comments to avoid citation. We firmly reject this:
Our responses are scientifically justified, not evasive. Citation decisions are based on relevance, not personal bias.